crystallography/organic chemistry/materials science

thermodynamics, DCBNT, energetic ionic salt, solubility

**Author for correspondence:**
Hongzhen Li
e-mail: hongzhenli@caep.cn

This article has been edited by the Royal Society of Chemistry, including the commissioning, peer review process and editorial aspects up to the point of acceptance.

# Solubility of dicarbohydrazide bis[3-(5-nitroimino-1,2,4-triazole)] in common pure solvents and binary solvents at different temperatures

Jianrong Ren[1,2], Dong Chen[1], Yanwu Yu[2] and Hongzhen Li[1]

[1]Institute of Chemical Materials, China Academy of Engineering Physics, Mianyang, Sichuan 621900, People's Republic of China
[2]College of Environment and Safety Engineering, North University of China, Taiyuan 030051, People's Republic of China

JR, 0000-0001-7706-5345; HL, 0000-0002-8924-3417

The solubility of dicarbohydrazide bis[3-(5-nitroimino-1,2,4-triazole)] (DCBNT) was first measured under the different pure solvents and binary solvents by the dynamic method over the temperature range of 290–360 K at atmospheric pressure. Results in all the solvents were positively correlated with temperature, namely increased with increasing temperature. The experiment data were correlated by the Apelblat equation, the Yaws equation and the polynomial equation. The conclusion showed that these three models all agreed well with the experimental data. Simultaneously, the dissolution enthalpy, dissolution entropy and Gibbs free energy of DCBNT in different solvents were calculated from the solubility data by using the Apelblat model. The results indicate that the dissolution process of DCBNT in these solvents is driven by entropy, which provides theoretical guidance for further research on the crystallization of DCBNT.

## 1. Introduction

Solubility evaluation plays a significant role in the purification and separation process in the industry of chemical production. It is well known that the density, energy, safety and compatibility with other chemicals of explosives are closely related to their crystal purity, particle size and morphology. In particular, the particle morphology of explosives was found to have important impact on

**Figure 1.** Molecular structure of dicarbohydrazide bis[3-(5-nitroimino-1,2,4-triazole)].

its safety and energy performance [1]. Therefore, in order to obtain high-quality and high-performance crystals, it is very important to design a reliable crystallization process and optimize the crystallization conditions in solvents to control the crystallization quality. The solubility data of compounds are important to control and optimize the crystallization process, since it will determine the selection of the crystallization method and the crystallization solvents [2–6]. On the other hand, thermodynamic parameters (dissolution enthalpy and entropy) can provide considerable information about the dissolving process of compounds in solvents, such as the endothermic or exothermic, entropy-driven and enthalpy-driven processes [7].

Nowadays, as alternatives to high-performance energetic materials, energetic ionic salts (EISs) have attracted increasing attention [8], especially for their lower vapour pressures, higher positive heats of formation, better thermal stability and higher densities than the atomically similar non-ionic compounds [8–10]. Dicarbohydrazide bis[3-(5-nitroimino-1,2,4-triazole)] (DCBNT) [11] (figure 1) is a novel EIS, with a moderate density of 1.780 g cm$^{-3}$, a high detonation velocity of 9234.87 m s$^{-1}$ and a detonation pressure of 31.73 GPa, which is calculated by EXPLO5 v. 6.02. Besides, DCBNT exhibits good thermal stability, as the decomposition peak temperature is over 230°C. Its impact sensitivity is greater than 40 J, and the friction sensitivity is 216 N. The high thermal stability, low sensitivity towards impact and friction as well as the good detonation properties make DCBNT a potential kind of low-sensitive and high-energetic explosive [12].

In this study, we tested the solubility of DCBNT in 12 commonly used solvents: water (H$_2$O), dimethyl sulfoxide (DMSO), N,N-diethylformamide (DEF), N,N-dimethylformamide (DMF), 1,4-butyrolactone (BL), methanol, ethanol, acetone, trichloromethane, dioxane, acetonitrile and ethyl acetate, and five binary solvents (volume ratio = 1 : 1), at atmospheric pressure using a polythermal method [13,14] with the CrystalSCAN system. The experimental solubility data were correlated by the modified Apelblat model, the Yaws model and the polynomial model. The thermodynamic magnitudes, such as the dissolution enthalpy, dissolution entropy and molar Gibbs free energy, were then obtained from the solubility data. The driving force of the process was determined by enthalpy–entropy compensation analysis [15].

# 2. Experimental

## 2.1. Materials

DCBNT [11] was synthesized by our research group according to Shreeve and co-workers [16]. The purity of DCBNT, 99.27%, was determined by high-performance liquid chromatography [17]. Distilled water was prepared in our laboratory and used throughout. All reagents were purchased commercially and used without further purification. The $^1$H NMR and $^{13}$C NMR are shown in figures 2 and 3, respectively.

## 2.2. Apparatus

The solubility data of DCBNT were measured by the dynamic method and collected by the CrystalSCAN system (E1320, HEL Ltd, UK; figure 4). The mass of DCBNT was weighed using an analytical balance (CP225D, Sartorius, Germany) with an accuracy of 10$^{-4}$ g. Circulating oil solution from a thermostat

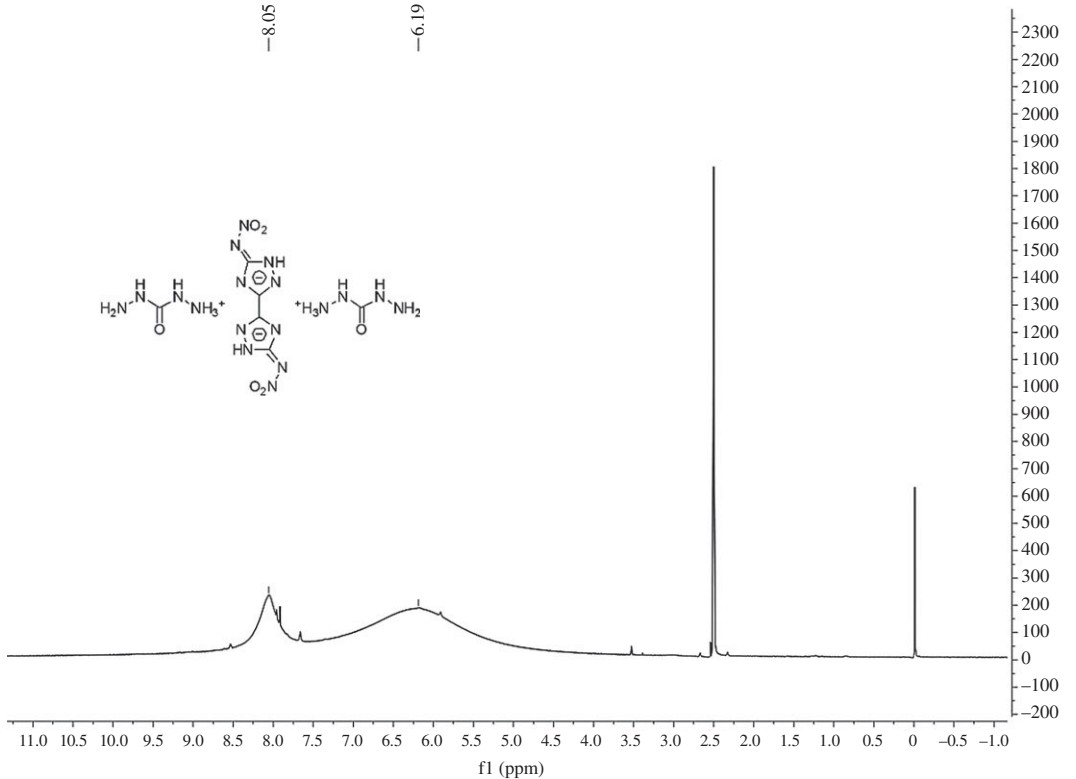

**Figure 2.** ¹H NMR spectrum of DCBNT.

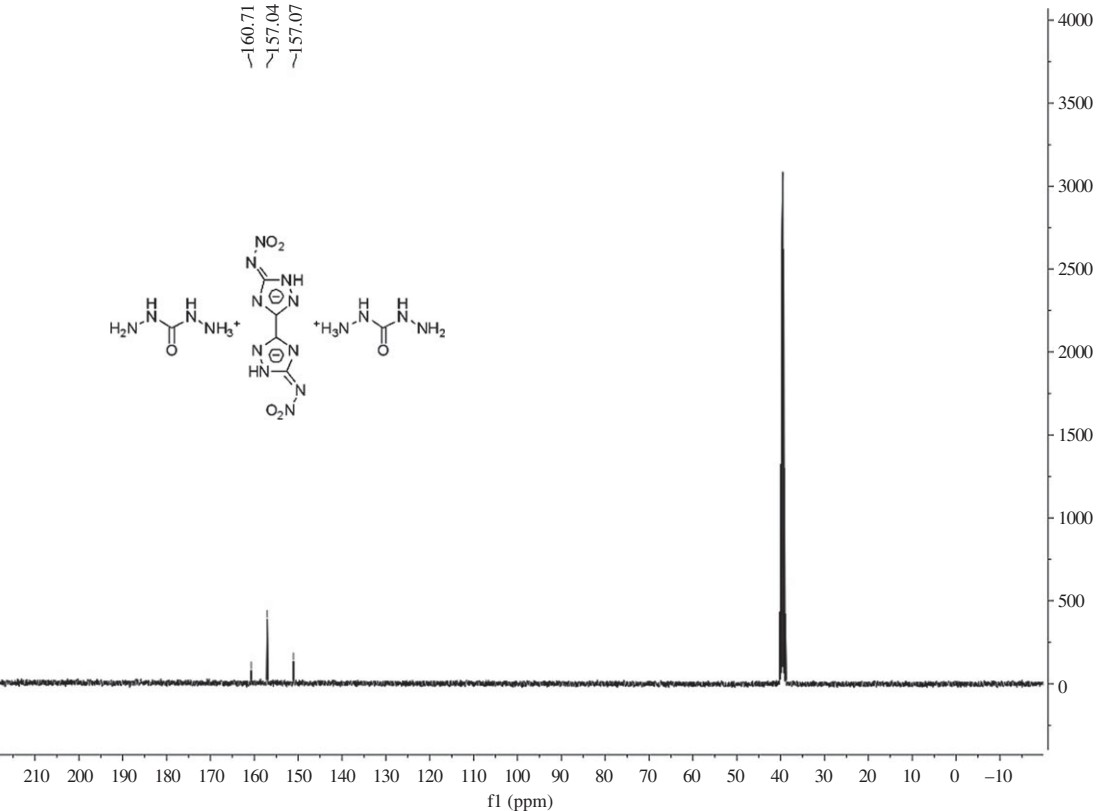

**Figure 3.** ¹³C NMR spectrum of DCBNT.

(Huber CC1-505wl vpc55, Germany) used with an uncertainty of $u(T) = 0.01$ K controlled the temperature of the mixture. ¹H and ¹³C spectra were recorded on a 400 MHz (Bruker AVANCE 400) or 600 MHz (Bruker AVANCE 600) nuclear magnetic resonance spectrometer.

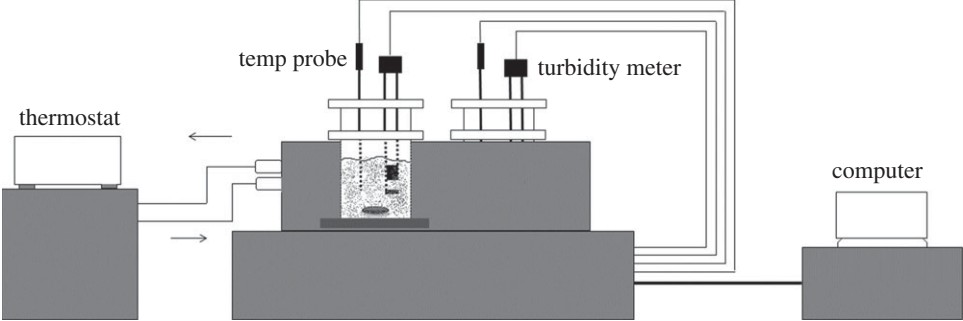

**Figure 4.** Schematic of the HEL CrystalSCAN system for solubility measurement.

## 2.3. Solubility determination

The solubility of DCBNT in all the solvents was tested by the dynamic method with a turbidity explorer. A known amount of DCBNT was added to an appropriate glass vial with 60 ml of solvent, the solution was then slowly heated at a specific speed and kept stirred, and the dissolved ability was judged by the turbidity curve. The heating rate was 0.2 K min$^{-1}$ and the stirring rate was 500 r.p.m. With temperature increasing, the turbidity changed gradually. When turbidity reached its minimum and remained unchanged for a long time, representing a full dissolution, this dissolution temperature was recorded as $T$. In order to reduce the deviation, each experiment was performed three times, and the average of three measurements was determined as the final value. The estimated relative standard uncertainty of the temperature was less than 0.003. The mole fraction solubility ($x$) of DCBNT in different pure solvents can be calculated by the following equation [18]:

$$x = \frac{m_1/M_1}{m_1/M_1 + m_2/M_2},$$

(2.1)

where $M_1$ and $M_2$ are the molecular masses of DCBNT and solvent, respectively; $m_1$ and $m_2$ represent the corresponding mass of DCBNT and solvent, respectively.

The calculation method for the mole fraction solubility ($x$) of DCBNT in binary solvents is the same as that of DBNT in pure solvent [19].

$$x = \frac{m_1/M_1}{m_1/M_1 + m_3/M_3 + m_4/M_4},$$

(2.2)

where $M_1$, $M_3$ and $M_4$, and $m_1$, $m_3$ and $m_4$ present the molecular masses and the masses of DCBNT, organic solvent and water, respectively.

# 3. Solubility models

All the solubility data obtained from pure solvents and binary solvents at different temperatures were correlated by three models: modified Apelblat model [20], Yaws model [21,22] and polynomial model, which were widely used.

## 3.1 Modified Apelblat model

The relationship between mole fraction solubility and temperature can be described by the Apelblat model. The expression is shown in the following equation:

$$\ln x = A_1 + \frac{B_1}{T} + C_1 \ln T,$$

(3.1)

where $x$ is the mole fraction solubility of DCBNT and $T$ is the absolute temperature (K). $A_1$, $B_1$ and $C_1$ are the empirical model parameters. They can be obtained to fit the experimental data by a nonlinear least-squares method [23].

## 3.2 Yaws model

For the Yaws model, the relationship between mole fraction solubility and temperature can be described as follows:

$$\ln x = A_2 + \frac{B_2}{T} + \frac{C_2}{(T)^2},$$
(3.2)

where $x$ is the mole fraction solubility of DCBNT; $T$ is the absolute temperature (K) and $A_2$, $B_2$ and $C_2$ are the empirical parameters of the model. They can be obtained to fit the experimental data by the nonlinear least-squares method.

## 3.3. Polynomial model

The relationship between mole fraction solubility of DCBNT and temperature was also correlated with the polynomial model. The specific expressions are as follows:

$$x = A_3 + B_3 T + C_3 T^2,$$
(3.3)

where $x$ is the mole fraction solubility of DCBNT; $T$ is the absolute temperature (K) and $A_3$, $B_3$ and $C_3$ are the empirical parameters of the model.

# 4. Results and discussion

## 4.1. Solubility data

It was found through experiments that the DCBNT is almost insoluble in most solvents, including DMF, methanol, ethanol, acetone, chloroform, dioxane, acetonitrile and ethyl acetate. On the other hand, DCBNT has better solubility in DMSO, $H_2O$, DEF and BL at temperatures from 290 to 360 K, and they are listed in table 1 and shown in figures 5–7. It can be found that the solubility of DCBNT in these selected pure solvents increased with increasing temperature. The solubility of DCBNT in DMSO is much higher than that in the other three solvents. Moreover, the order of DCBNT solubility in different solvents is: DMSO > DEF > $H_2O$ > BL, by further comparing the four sets of data, and it can also be seen that the mole fraction solubility of DCBNT in DMSO is nearly 100 times higher than that in $H_2O$. According to the principle of 'like dissolves like' [24,25], the solubility of DCBNT in $H_2O$ should be better than that in DMSO, so the solubility of DCBNT may not only depend upon the solvent polarity but also upon other factors. Although the solubility of DCBNT in $H_2O$, DEF and BL is not so good as in DMSO, the solubility curve changes obviously with temperature, so it can also be used as an alternative solvent for cooling crystallization of DCBNT.

The comparison between the calculated and experimental values is shown in table 1. The relative deviation (RD) is given in table 1. The regression parameters of each model are given in table 2. In addition, we calculated the relative average deviation (RAD) and root-mean-square deviation (RMSD), which are important for evaluating the applicability and accuracy of the models used in this study. RD is shown in the following equation:

$$\text{RD} = \frac{x_i^{\text{exp}} - x_i^{\text{cal}}}{x_i^{\text{exp}}}.$$
(4.1)

The RAD is described as follows:

$$\text{RAD} = \frac{1}{N} \sum_{i=1}^{N} \left| \frac{x_i^{\text{exp}} - x_i^{\text{cal}}}{x_i^{\text{exp}}} \right|.$$
(4.2)

The RMSD is defined as follows:

$$\text{RMSD} = \sqrt{\frac{\sum_{i=1}^{N} (x_i^{\text{exp}} - x_i^{\text{cal}})^2}{N}}.$$
(4.3)

**Table 1.** Mole fraction solubility $x$ of DCBNT in pure solvents at different temperatures under 101 kPa[a].

| $T$ (K)[b] | $1000x$[c] | Apelblat model | | polynomial model | | Yaws model | |
|---|---|---|---|---|---|---|---|
| | | $1000x^{cal}$ | RD | $1000x^{cal}$ | RD | $1000x^{cal}$ | RD |
| DMSO | | | | | | | |
| 296.9 | 1.06 | 1.95 | −0.840 | 0.88 | 0.170 | 1.89 | −0.783 |
| 301.2 | 2.52 | 2.75 | −0.091 | 2.45 | 0.028 | 2.71 | −0.075 |
| 307.2 | 4.22 | 4.16 | 0.014 | 4.58 | −0.085 | 4.15 | 0.017 |
| 312.2 | 6.02 | 5.56 | 0.076 | 6.19 | −0.028 | 5.59 | 0.071 |
| 317.9 | 7.85 | 7.35 | 0.064 | 7.89 | −0.005 | 7.40 | 0.057 |
| 325.2 | 9.61 | 9.71 | −0.104 | 9.81 | −0.021 | 9.75 | −0.015 |
| 333 | 11.82 | 11.98 | −0.014 | 11.56 | 0.022 | 11.98 | −0.014 |
| 340.9 | 13.38 | 12.43 | 0.071 | 12.98 | 0.030 | 13.54 | −0.012 |
| 348.1 | 13.99 | 14.23 | −0.017 | 14.00 | −0.001 | 14.2 | −0.015 |
| 354.6 | 14.42 | 14.09 | 0.023 | 14.68 | −0.018 | 14.14 | 0.019 |
| H$_2$O | | | | | | | |
| 291.3 | 0.010 | 0.017 | −0.700 | 0.015 | −0.500 | 0.017 | −0.700 |
| 299.4 | 0.030 | 0.032 | −0.067 | 0.029 | 0.033 | 0.032 | −0.067 |
| 303.7 | 0.040 | 0.044 | −0.100 | 0.041 | −0.025 | 0.044 | −0.100 |
| 307 | 0.060 | 0.055 | 0.083 | 0.053 | 0.117 | 0.055 | 0.083 |
| 314.1 | 0.091 | 0.086 | 0.055 | 0.087 | 0.044 | 0.086 | 0.055 |
| 324.1 | 0.151 | 0.148 | 0.020 | 0.152 | −0.001 | 0.148 | 0.020 |
| 329.7 | 0.192 | 0.193 | −0.005 | 0.196 | −0.021 | 0.193 | −0.005 |
| 333 | 0.222 | 0.224 | −0.010 | 0.226 | −0.018 | 0.224 | −0.009 |
| 337.3 | 0.252 | 0.267 | −0.060 | 0.268 | −0.063 | 0.266 | −0.056 |
| 341.5 | 0.323 | 0.312 | 0.034 | 0.312 | 0.034 | 0.312 | 0.034 |
| 346.5 | 0.374 | 0.372 | 0.005 | 0.370 | 0.011 | 0.371 | 0.008 |
| 349.3 | 0.404 | 0.406 | −0.005 | 0.404 | 0 | 0.406 | −0.005 |
| DEF | | | | | | | |
| 296 | 0.042 | 0.052 | −0.238 | 0.034 | 0.190 | 0.050 | −0.190 |
| 302.6 | 0.083 | 0.083 | 0 | 0.082 | 0.012 | 0.082 | 0.012 |
| 309.7 | 0.125 | 0.127 | −0.016 | 0.137 | −0.096 | 0.127 | −0.016 |
| 314.1 | 0.166 | 0.161 | 0.030 | 0.171 | −0.030 | 0.161 | 0.030 |
| 321 | 0.208 | 0.221 | −0.063 | 0.227 | −0.091 | 0.222 | −0.067 |
| 325.5 | 0.291 | 0.264 | 0.093 | 0.266 | 0.086 | 0.265 | 0.089 |
| 332.8 | 0.332 | 0.335 | −0.009 | 0.329 | 0.009 | 0.336 | −0.012 |
| 339 | 0.374 | 0.394 | −0.053 | 0.385 | −0.029 | 0.394 | −0.053 |
| 340.8 | 0.416 | 0.411 | 0.012 | 0.401 | 0.036 | 0.410 | 0.014 |
| 346.7 | 0.457 | 0.458 | −0.02 | 0.456 | 0.002 | 0.458 | −0.002 |
| 352 | 0.499 | 0.493 | 0.012 | 0.507 | −0.016 | 0.494 | 0.010 |
| BL | | | | | | | |
| 301.2 | 0.033 | 0.044 | −0.333 | 0.038 | −0.152 | 0.044 | −0.333 |
| 311.4 | 0.067 | 0.066 | 0.015 | 0.065 | 0.030 | 0.065 | 0.030 |
| 320.2 | 0.100 | 0.089 | 0.110 | 0.091 | 0.09 | 0.088 | 0.120 |

(*Continued.*)

| $T$ (K)[b] | $1000x$[c] | Apelblat model | | polynomial model | | Yaws model | |
| | | $1000x^{cal}$ | RD | $1000x^{cal}$ | RD | $1000x^{cal}$ | RD |
|---|---|---|---|---|---|---|---|
| 333 | 0.133 | 0.131 | 0.015 | 0.134 | −0.008 | 0.131 | 0.015 |
| 343 | 0.167 | 0.170 | −0.018 | 0.172 | −0.030 | 0.171 | −0.023 |
| 353.2 | 0.200 | 0.215 | −0.075 | 0.215 | −0.075 | 0.215 | −0.075 |
| 358 | 0.250 | 0.238 | 0.048 | 0.236 | 0.056 | 0.238 | 0.048 |

[a]Standard uncertainties $u$ are $u(T) = 0.01$ K, $u(P) = 3$ kPa.

[b]The estimated relative standard uncertainty of the temperature is $ur(T) = 0.003$.

[c]$x$ is the experimental solubility data of DCBNT.

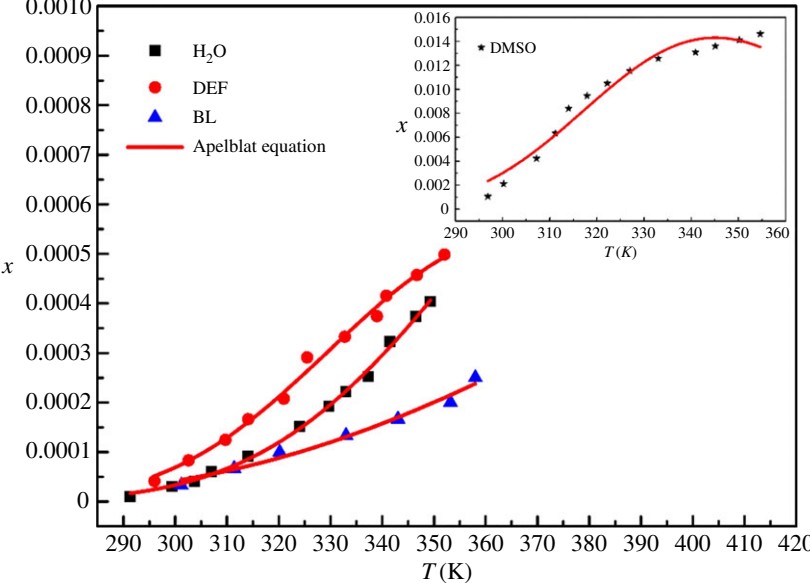

**Figure 5.** Mole fraction solubility $x$ of DCBNT in different solvents: (⋆) DMSO; (■) $H_2O$; (●) DEF; (▲) BL. The line is the best fit of the experimental data calculated with the Apelblat equation.

In equations (4.1)–(4.3), $x_i^{exp}$ and $x_i^{cal}$ represent the experimental and computational values of molar fractional solubility of DCBNT, respectively. $N$ represents the number of points measured in the experiment.

As can be seen from figures 5–7, the experimental data are basically consistent with the empirical equation data, and the experimental data are evenly distributed near the fitting line. The closer the $R^2$ value is to 1, the higher the reference value of the empirical equation. From tables 1 and 2, we can find that the values of correlation coefficient ($R^2$) are all close to 1, which indicates that the values obtained by the three models are in good agreement with the experimental values, especially in DMSO, DEF and $H_2O$. In addition, we also find that the Apelblat model is better than the polynomial model and the Yaws model in correlating solubility data in DMSO and BL. For DEF and $H_2O$, the Yaws model is better than the Apelblat equation and the polynomial model. Moreover, the RADs and RMSDs obtained by fitting the solubility data of DCBNT in four pure solvents by the three models are not very different. In terms of RMSD, it will be found that the values of DMSO, $H_2O$, DEF and BL ($2.31 \times 10^{-4}$, $6.57 \times 10^{-6}$, $1.22 \times 10^{-5}$ and $8.92 \times 10^{-6}$) correlated with the polynomial model are slightly better than those fitted by the Apelblat model ($4.9 \times 10^{-4}$, $6.37 \times 10^{-6}$, $1.17 \times 10^{-5}$ and $9.45 \times 10^{-6}$) and the Yaws model ($3.62 \times 10^{-4}$, $6.21 \times 10^{-6}$, $1.15 \times 10^{-5}$ and $9.70 \times 10^{-6}$), which shows that the calculated values obtained by the polynomial method are less deviated from the experimental values. In sum, all three models are suitable for describing the solubility of DCBNT in the selected pure solvents.

In the crystallization process, when the solubility of compounds in pure solvents is low, recrystallization with mixed solvents is a common method. The solubility of DCBNT in different binary solvents was also tested in the range of 290–360 K. The results made clear that the solubility of DCBNT in acetone + $H_2O$ is

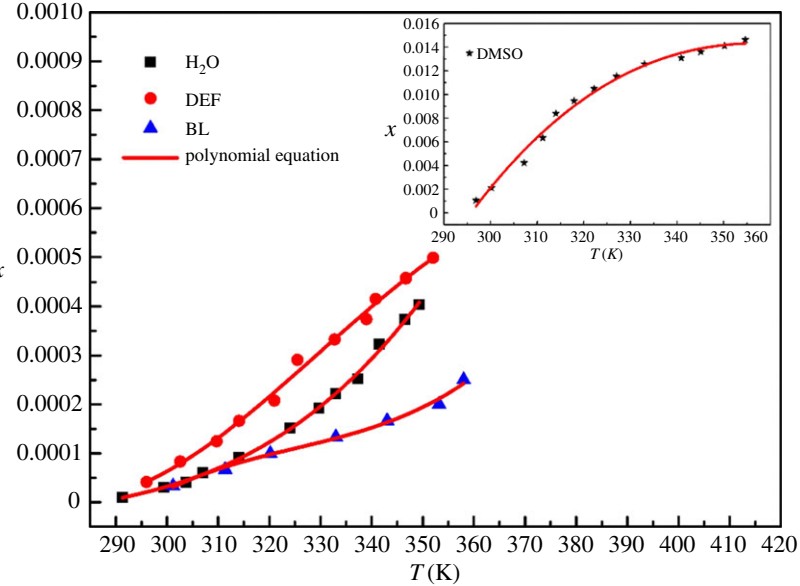

**Figure 6.** Mole fraction solubility $x$ of DCBNT in different solvents: (★) DMSO; (■) H$_2$O; (●) DEF; (▲) BL. The line is the best fit of the experimental data calculated with the polynomial equation.

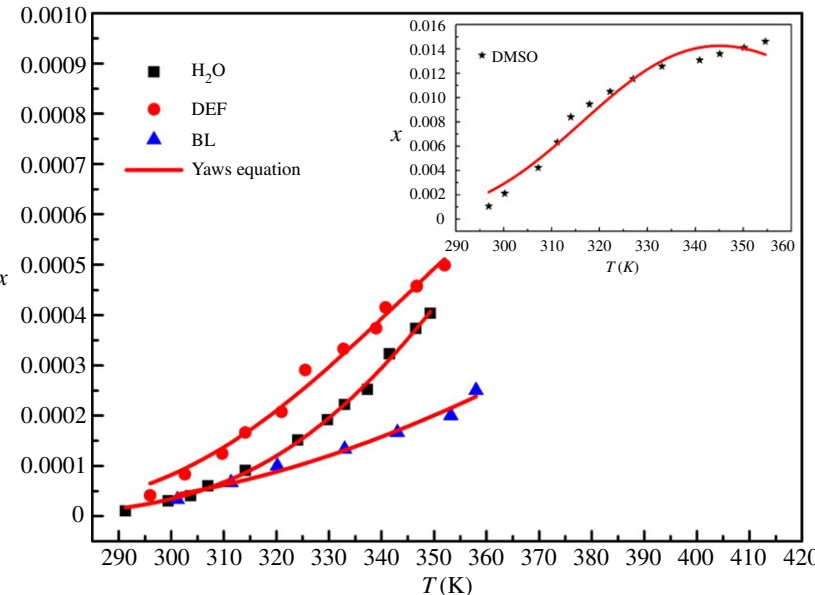

**Figure 7.** Mole fraction solubility $x$ of DCBNT in different solvents: (★) DMSO; (■) H$_2$O; (●) DEF; (▲) BL. The line is the best fit of the experimental data calculated with the Yaws equation.

abnormal (the experimental data fluctuate greatly, it is probably because acetone evaporates too quickly) and almost insoluble in methanol + H$_2$O and ethanol + H$_2$O. Therefore, the available mixed solvents include BL + H$_2$O, DMSO + H$_2$O, DEF + H$_2$O, DMF + H$_2$O and ACN + H$_2$O. The results are shown in table 3. The solubility data in these five binary solvents (volume ratio = 1 : 1) are also correlated by the Apelblat model equation, the Yaws model and the polynomial model equation (equations (3.1)–(3.3)), and the values of the parameters are listed in table 4. The RADs and RMSDs are also given in table 4. Figures 8–10 are curves of the mole fraction solubility $x$ of DCBNT in five binary solvents fitting by three model equations, respectively. In terms of solubility for DCBNT in five binary solvents, they all increased with increasing temperature, which indicates that the dissolution process is endothermic. Compared with these three binary solvents, the solubility of DCBNT in ACN + H$_2$O increased slowly with temperature, but in BL + H$_2$O the solubility was fastest. The order of DCBNT solubility in binary solvents is BL + H$_2$O > DMSO + H$_2$O > DEF + H$_2$O > DMF + H$_2$O > ACN + H$_2$O.

**Table 2.** Regression parameters, RAD and RMSD for the solubility of DCBNT in pure solvents with different models at pressure $p = 101$ kPa[a].

| equation | solvent | A[b] | B[b] | C[b] | $R^{2}$[c] | RAD[d] | RMSD[e] |
|---|---|---|---|---|---|---|---|
| Apelblat model | DMSO | 947.879 | −48 593.720 | −138.836 | 0.9907 | 0.131 | $4.9 \times 10^{-4}$ |
|  | $H_2O$ | 394.398 | −23 740.123 | −57.077 | 0.99733 | 0.095 | $6.37 \times 10^{-6}$ |
|  | DEF | 632.005 | −34 044.067 | −92.588 | 0.99225 | 0.050 | $1.17 \times 10^{-5}$ |
|  | BL | 134.809 | −9677.480 | −19.746 | 0.97239 | 0.088 | $9.45 \times 10^{-6}$ |
| polynomial model | DMSO | −0.349 | 0.002 | $-2.642 \times 10^{-6}$ | 0.9967 | 0.041 | $2.31 \times 10^{-4}$ |
|  | $H_2O$ | 0.008 | $-5.837 \times 10^{-5}$ | $1.016 \times 10^{-7}$ | 0.9970 | 0.072 | $6.57 \times 10^{-6}$ |
|  | DEF | $-1.025 \times 10^{-4}$ | $-6.254 \times 10^{-6}$ | $2.268 \times 10^{-8}$ | 0.9915 | 0.010 | $1.22 \times 10^{-5}$ |
|  | BL | $9.947 \times 10^{-4}$ | $-8.780 \times 10^{-6}$ | $1.861 \times 10^{-8}$ | 0.9763 | 0.063 | $8.92 \times 10^{-6}$ |
| Yaws model | DMSO | −65.849 | 43 203.612 | −7.576 | 0.9922 | 0.010 | $3.62 \times 10^{-4}$ |
|  | $H_2O$ | −22.062 | 13 804.300 | −3.083 | 0.9974 | 0.095 | $6.21 \times 10^{-6}$ |
|  | DEF | −43.986 | 27 144.441 | −5.048 | 0.9926 | 0.004 | $1.15 \times 10^{-5}$ |
|  | BL | −10.326 | 4106.828 | −1.84 | 0.9729 | 0.092 | $9.70 \times 10^{-6}$ |

[a]The standard uncertainty is $u(P) = 3$ kPa.
[b]A, B and C are parameters of different models.
[c]$R^2$ stands for correlation coefficient.
[d]RAD is the average relative deviation.
[e]RMSD represents the root-mean-square deviation.

**Table 3.** Mole fraction solubility $x$ of DCBNT in the binary solvent mixtures (volume ratio = 1 : 1) at different temperatures under 101 kPa[a].

| $T$ (K)[b] | $1000x$[c] | Apelblat model | | polynomial model | | Yaws model | |
|---|---|---|---|---|---|---|---|
| | | $1000x^{cal}$ | RD | $1000x^{cal}$ | RD | $1000x^{cal}$ | RD |
| DMSO + $H_2O$ | | | | | | | |
| 300.2 | 0.082 | 0.102 | −0.246 | 0.103 | −0.256 | 0.102 | −0.244 |
| 310.5 | 0.164 | 0.160 | 0.020 | 0.147 | 0.104 | 0.160 | 0.024 |
| 315 | 0.205 | 0.196 | 0.046 | 0.184 | 0.102 | 0.195 | 0.049 |
| 320 | 0.246 | 0.243 | 0.012 | 0.237 | 0.037 | 0.242 | 0.016 |
| 326.5 | 0.328 | 0.322 | 0.018 | 0.327 | 0.003 | 0.323 | 0.015 |
| 330.2 | 0.39 | 0.378 | 0.032 | 0.388 | 0.005 | 0.378 | 0.031 |
| 334.6 | 0.451 | 0.456 | −0.011 | 0.470 | −0.042 | 0.456 | −0.011 |
| 337.6 | 0.512 | 0.519 | −0.012 | 0.532 | −0.039 | 0.519 | −0.014 |
| 340.6 | 0.574 | 0.591 | −0.025 | 0.599 | −0.044 | 0.590 | −0.028 |
| 345 | 0.697 | 0.710 | −0.017 | 0.705 | −0.011 | 0.711 | −0.020 |
| 347.9 | 0.82 | 0.801 | 0.023 | 0.780 | 0.049 | 0.802 | 0.022 |
| ACN + $H_2O$ | | | | | | | |
| 298.8 | 0.041 | 0.047 | −0.140 | 0.046 | −0.122 | 0.047 | −0.136 |
| 306.5 | 0.062 | 0.063 | −0.013 | 0.062 | 0 | 0.063 | −0.013 |
| 309.5 | 0.077 | 0.070 | 0.087 | 0.070 | 0.091 | 0.070 | 0.086 |
| 313 | 0.082 | 0.080 | 0.023 | 0.080 | 0.024 | 0.080 | 0.022 |
| 322.1 | 0.113 | 0.112 | 0.012 | 0.114 | −0.009 | 0.112 | 0.009 |
| 331.9 | 0.154 | 0.158 | −0.025 | 0.161 | −0.045 | 0.158 | −0.028 |
| 336.7 | 0.185 | 0.187 | −0.008 | 0.188 | −0.016 | 0.187 | −0.012 |
| 342 | 0.226 | 0.224 | 0.012 | 0.220 | 0.027 | 0.223 | 0.012 |
| BL + $H_2O$ | | | | | | | |
| 298 | 0.125 | 0.078 | 0.376 | 0.122 | 0.024 | 0.079 | 0.367 |
| 306 | 0.167 | 0.139 | 0.168 | 0.153 | 0.084 | 0.139 | 0.170 |
| 311 | 0.208 | 0.188 | 0.096 | 0.194 | 0.067 | 0.190 | 0.088 |
| 317 | 0.25 | 0.267 | −0.068 | 0.264 | −0.056 | 0.269 | −0.072 |
| 323.4 | 0.292 | 0.373 | −0.277 | 0.365 | −0.25 | 0.374 | −0.279 |
| 329.7 | 0.5 | 0.502 | 0.043 | 0.491 | 0.018 | 0.500 | −0.021 |
| 332.9 | 0.542 | 0.577 | −0.004 | 0.564 | −0.041 | 0.575 | −0.061 |
| 337 | 0.733 | 0.682 | −0.065 | 0.670 | 0.086 | 0.680 | 0.074 |
| 346 | 1 | 0.943 | 0.057 | 0.937 | 0.063 | 0.941 | 0.059 |
| 352.2 | 1.108 | 1.145 | −0.033 | 1.154 | −0.042 | 1.150 | −0.033 |
| 353 | 1.167 | 1.171 | −0.003 | 1.182 | −0.013 | 1.170 | −0.005 |
| DEF + $H_2O$ | | | | | | | |
| 299.4 | 0.034 | 0.080 | −1.353 | 0.065 | −0.912 | 0.079 | −1.324 |
| 302.7 | 0.059 | 0.094 | −0.593 | 0.083 | −0.407 | 0.094 | −0.593 |
| 303.9 | 0.118 | 0.100 | 0.110 | 0.091 | 0.229 | 0.100 | 0.153 |
| 308.7 | 0.153 | 0.126 | 0.153 | 0.122 | 0.203 | 0.126 | 0.176 |
| 314.2 | 0.189 | 0.162 | 0.176 | 0.165 | 0.127 | 0.162 | 0.143 |
| 319.5 | 0.224 | 0.204 | 0.143 | 0.213 | 0.049 | 0.204 | 0.089 |

(*Continued.*)

**Table 3.** (*Continued.*)

| T (K)[b] | 1000x[c] | Apelblat model | | polynomial model | | Yaws model | |
|---|---|---|---|---|---|---|---|
| | | 1000x$^{cal}$ | RD | 1000x$^{cal}$ | RD | 1000x$^{cal}$ | RD |
| 323.3 | 0.248 | 0.239 | 0.089 | 0.251 | −0.012 | 0.240 | 0.032 |
| 327.7 | 0.283 | 0.286 | 0.036 | 0.299 | −0.057 | 0.287 | −0.014 |
| 334.6 | 0.354 | 0.373 | −0.011 | 0.383 | −0.082 | 0.374 | −0.056 |
| 338.8 | 0.413 | 0.436 | −0.054 | 0.439 | −0.063 | 0.436 | −0.056 |
| 341.8 | 0.472 | 0.485 | −0.056 | 0.482 | −0.021 | 0.485 | −0.028 |
| 346.2 | 0.59 | 0.565 | 0.042 | 0.548 | 0.071 | 0.564 | 0.044 |
| DMF + H$_2$O | | | | | | | |
| 300 | 0.033 | 0.073 | −1.212 | 0.058 | −0.758 | 0.072 | −1.183 |
| 304 | 0.067 | 0.089 | −0.328 | 0.078 | −0.164 | 0.088 | −0.318 |
| 307.4 | 0.117 | 0.105 | 0.103 | 0.098 | 0.162 | 0.105 | 0.109 |
| 312.9 | 0.167 | 0.135 | 0.192 | 0.134 | 0.198 | 0.135 | 0.193 |
| 323 | 0.233 | 0.208 | 0.107 | 0.216 | 0.073 | 0.208 | 0.108 |
| 330 | 0.283 | 0.273 | 0.035 | 0.285 | −0.007 | 0.274 | 0.033 |
| 336.8 | 0.333 | 0.350 | −0.051 | 0.362 | −0.087 | 0.351 | −0.053 |
| 339.2 | 0.383 | 0.382 | 0.003 | 0.366 | 0.044 | 0.382 | 0.004 |
| 345.3 | 0.433 | 0.468 | −0.081 | 0.391 | 0.097 | 0.468 | −0.080 |
| 351.3 | 0.549 | 0.566 | −0.031 | 0.557 | −0.015 | 0.566 | −0.030 |
| 352 | 0.616 | 0.578 | 0.062 | 0.567 | 0.080 | 0.577 | 0.063 |

[a]Standard uncertainties $u$ are $u(T) = 0.01$ K, $u(P) = 3$ kPa.
[b]The estimated relative standard uncertainty of the temperature is $ur(T) = 0.003$.
[c]$x$ is the experimental solubility data of DCBNT.

From tables 3 and 4, we can also find that the values of correlation coefficient ($R^2$) are all close to 1, which shows that the values obtained by three models agreed well with the experimental values as well as the RDs. However, the polynomial model is superior to the Apelblat model and the Yaws model in the correlation results of DMSO + H$_2$O, DEF + H$_2$O, DMF + H$_2$O and ACN + H$_2$O. At the same time, the correlation results of BL + H$_2$O show that the Yaws model is superior to the Apelblat equation and the polynomial model. Besides, the values of RADs and RMSDs of the polynomial model, the Apelblat model and the Yaws model are basically consistent. From RMDS, the values associated with the Apelblat model in DMSO + H$_2$O, ACN + H$_2$O, BL + H$_2$O, DEF + H$_2$O and DMF + H$_2$O ($1.20 \times 10^{-5}$, $3.68 \times 10^{-6}$, $4.12 \times 10^{-5}$, $2.47 \times 10^{-5}$ and $2.56 \times 10^{-5}$) were better than those associated with the polynomial model ($1.98 \times 10^{-5}$, $4.65 \times 10^{-6}$, $3.90 \times 10^{-5}$, $2.51 \times 10^{-5}$ and $2.67 \times 10^{-5}$) and the Yaws model ($1.19 \times 10^{-5}$, $3.70 \times 10^{-6}$, $4.21 \times 10^{-5}$, $2.46 \times 10^{-5}$ and $2.56 \times 10^{-5}$), indicating that the deviation between the calculated value and the experimental value obtained by the Apelblat model is smaller. In conclusion, the Apelblat model, the Yaws model and the polynomial model can accurately correlate the solubility of DCBNT in binary solvents composed of organic solvents and water. Therefore, we believe that these three models can be used to correlate the solubility data of DCBNT in further study of DCBNT.

In sum, the solubility of DCBNT in all solvents increased with increasing temperature, showing that the solubility of DCBNT in various solvents is closely related to temperature. Likewise, the composition of solvent has a great influence on the solubility of DCBNT. These results provide a theoretical basis for the thermodynamic analysis of the dissolution process.

## 4.2. Thermodynamic properties of DCBNT in solution

The thermodynamic properties for DCBNT in different solvents were described through the standard dissolution enthalpy, standard dissolution entropy and Gibbs free energy, which were calculated according to the modified Apelblat model equation [7,26]. The equation for standard molar

**Table 4.** Regressed parameters, RAD and RMSD for the solubility of DCBNT in binary (volume ratio = 1 : 1) solvents with three models at pressure $P = 101$ kPa[a].

| equation | solvent | A[b] | B[b] | C[b] | $R^{2c}$ | RAD[d] | RMSD[e] |
|---|---|---|---|---|---|---|---|
| Apelblat model | DMSO + H$_2$O | −155.814 | 3179.633 | 23.846 | 0.9966 | 0.042 | $1.20 \times 10^{-5}$ |
| | ACN + H$_2$O | −65.943 | −477.500 | 10.100 | 0.9949 | 0.040 | $3.68 \times 10^{-6}$ |
| | BL + H$_2$O | 381.593 | −23 020.226 | −55.080 | 0.9848 | 0.111 | $4.12 \times 10^{-5}$ |
| | DEF + H$_2$O | 71.139 | −7468.460 | −9.756 | 0.9717 | 0.228 | $2.47 \times 10^{-5}$ |
| | DMF + H$_2$O | 107.910 | −9153.307 | −15.240 | 0.9755 | 0.200 | $2.56 \times 10^{-5}$ |
| polynomial model | DMSO + H$_2$O | 23.695 | −0.159 | $2.67 \times 10^{-4}$ | 0.9901 | 0.063 | $1.98 \times 10^{-5}$ |
| | ACN + H$_2$O | 4.475 | −0.031 | $5.517 \times 10^{-5}$ | 0.9920 | 0.042 | $4.65 \times 10^{-6}$ |
| | BL + H$_2$O | 28.938 | −0.195 | $3.285 \times 10^{-4}$ | 0.9863 | 0.068 | $3.90 \times 10^{-5}$ |
| | DEF + H$_2$O | 8.561 | −0.062 | $1.117 \times 10^{-4}$ | 0.9706 | 0.186 | $2.51 \times 10^{-5}$ |
| | DMF + H$_2$O | 7.823 | −0.056 | $1.013 \times 10^{-4}$ | 0.9751 | 0.153 | $2.67 \times 10^{-5}$ |
| Yaws model | DMSO + H$_2$O | 24.505 | −12 122.423 | 1.225 | 0.9965 | 0.043 | $1.19 \times 10^{-5}$ |
| | ACN + H$_2$O | 13.944 | −6657.137 | 470 401.797 | 0.9948 | 0.040 | $3.70 \times 10^{-4}$ |
| | BL + H$_2$O | −12.560 | 12 645.584 | −2.879 | 0.9844 | 0.112 | $4.21 \times 10^{-5}$ |
| | DEF + H$_2$O | 5.707 | −294.364 | −650 820.894 | 0.9718 | 0.226 | $2.46 \times 10^{-5}$ |
| | DMF + H$_2$O | 2.638 | 1524.471 | −931 513.005 | 0.9756 | 0.198 | $2.56 \times 10^{-5}$ |

[a]The standard uncertainty is $u(P) = 3$ kPa.
[b]A, B and C refer to the parameters of these models.
[c]$R^2$ is the correlation coefficient.
[d]RAD is the average relative deviation.
[e]RMSD represents the root-mean-square deviation.

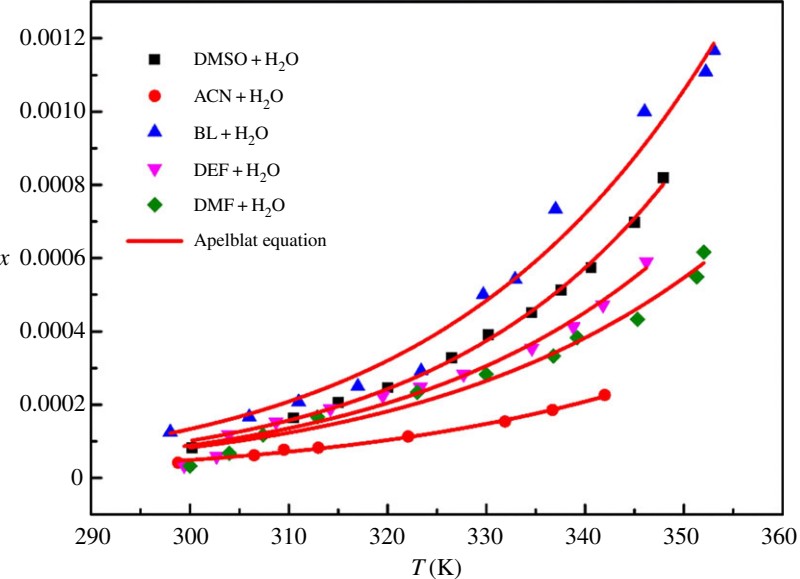

**Figure 8.** Mole fraction solubility x of DCBNT in binary solvents. The line is the best fit of the experimental data calculated with the Apelblat equation.

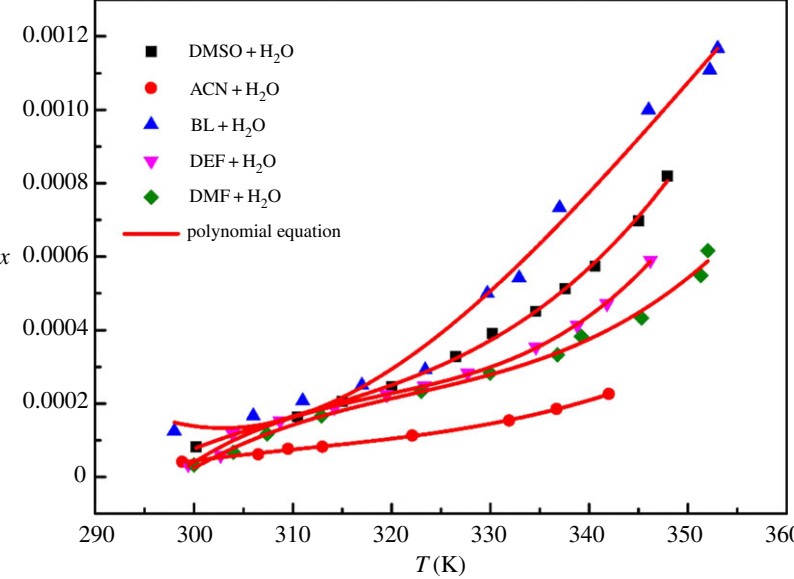

**Figure 9.** Mole fraction solubility x of DCBNT in binary solvents. The line is the best fit of the experimental data calculated with the polynomial equation.

dissolution enthalpy ($\Delta H_{sol}$) is as follows:

$$\Delta H_{sol} = -R \times \left( \frac{\partial \ln x}{\partial(1/T)} \right), \tag{4.4}$$

where $\Delta H_{sol}$ is the standard molar enthalpy dissolution; $R$ is the gas constant; $x$ is the mole fraction solubility of DCBNT; and $T$ is the solution temperature (K).

From equations (3.1) and (4.4), equation (4.5) can be obtained as follows:

$$\Delta H_{sol} = RT \left( C - \frac{B}{T} \right). \tag{4.5}$$

The equation of mole Gibbs free energy is shown as follows:

$$\Delta G_{sol} = -RT \ln x. \tag{4.6}$$

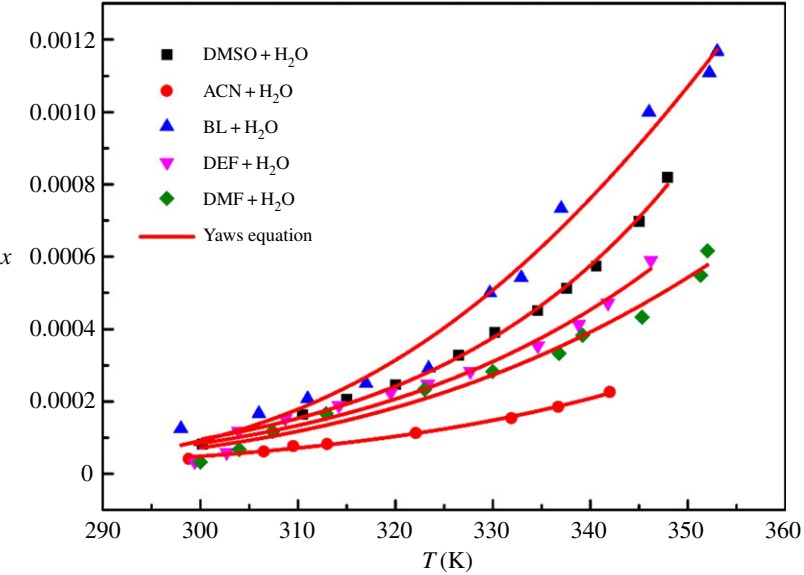

**Figure 10.** Mole fraction solubility $x$ of DCBNT in different binary solvents. The line is the best fit of the experimental data calculated with the Yaws equation.

The molar entropy of dissolution can be obtained through the standard molar dissolution enthalpy and mole Gibbs free energy, as shown in the following equation [27,28]:

$$\Delta S_{sol} - \frac{\Delta H_{sol} - \Delta G_{sol}}{T}. \tag{4.7}$$

The final functions were obtained as follows:

$$\Delta H_{sol} = RT\left(C - \frac{B}{T}\right), \tag{4.8}$$

$$\Delta S_{sol} = R(A + C + C\ln T) \tag{4.9}$$

and

$$\Delta G_{sol} = -RT\left(A + \frac{B}{T} + C\ln T\right), \tag{4.10}$$

where $A$, $B$ and $C$ are the parameters gained from the modified Apelblat model (tables 2 and 4). The mean temperature $T$ was defined by the following equation for minimizing the error propagation [29,30]:

$$T = \frac{N}{\sum (1/T_i)}, \tag{4.11}$$

where $N$ is the number of temperature points measured in the experiment.

The following equations are used to compare the relative contribution of enthalpy ($\%\zeta_H$) and entropy ($\%\zeta_{TS}$) to the dissolution of DCBNT:

$$\%\zeta_H = \frac{|\Delta H_{sol}|}{|\Delta H_{sol}| + |T\Delta S_{sol}|} \times 100 \tag{4.12}$$

and

$$\%\zeta_{TS} = \frac{|T\Delta S_{sol}|}{|\Delta H_{sol}| + |T\Delta S_{sol}|} \times 100. \tag{4.13}$$

The variables $\Delta H_{sol}$, $\Delta S_{sol}$, $\Delta G_{sol}$, $\%\zeta_H$ and $\%\zeta_{TS}$ were calculated from equations (4.4) to (4.13) and summarized in tables 5 and 6. $\Delta H_{sol}$ and $\Delta G_{sol}$ in pure and binary solvents are all positive, indicating that the dissolution process of DCBNT in all tested solvents is endothermic [31,32].

The results can be extracted from tables 5 and 6 that the enthalpy and the standard Gibbs free energy of DCBNT are positive in both studied pure solvent and binary solvents, indicating that the solution process of DCBNT in all of these solvents is endothermic. The values of $\Delta S_{sol}$ were positive, indicating

**Table 5.** Thermodynamic properties for the dissolution of DCBNT in pure solvents.

| solvents | $T$ | $\Delta H_{sol}$ (kJ mol$^{-1}$)[a] | $\Delta S_{sol}$ (J mol$^{-1}$ K$^{-1}$)[b] | $\Delta G_{sol}$ (kJ mol$^{-1}$)[c] | %$\zeta_H$[d] | %$\zeta_{TS}$[e] |
|---|---|---|---|---|---|---|
| DMSO | 322.6 | 31.64 | 58.77 | 12.68 | 62.53 | 37.47 |
| H$_2$O | 322.0 | 31.01 | 64.24 | 23.89 | 59.99 | 40.01 |
| DEF | 324.5 | 33.25 | 33.64 | 22.33 | 75.28 | 24.72 |
| BL | 330.2 | 26.25 | 4.51 | 24.76 | 94.63 | 5.37 |

[a]The solution enthalpy of DCBNT.
[b]The solution entropy of DCBNT.
[c]The Gibbs free energy of DCBNT dissolution in solution.
[d]The relative contributions of enthalpy to dissolution of DCBNT.
[e]The relative contributions of entropy to dissolution of DCBNT.

**Table 6.** Thermodynamic properties for the dissolution of DCBNT in binary solvents.

| solvents | $T$ | $\Delta H_{sol}$ (kJ mol$^{-1}$)[a] | $\Delta S_{sol}$ (J mol$^{-1}$ K$^{-1}$)[b] | $\Delta G_{sol}$ (kJ mol$^{-1}$)[c] | %$\zeta_H$[d] | %$\zeta_{TS}$[e] |
|---|---|---|---|---|---|---|
| DMSO + H$_2$O | 327.4 | 38.47 | 50.95 | 21.79 | 69.75 | 30.25 |
| ACN + H$_2$O | 319.4 | 30.79 | 19.94 | 24.42 | 82.86 | 17.14 |
| BL + H$_2$O | 326.9 | 41.69 | 63.34 | 20.98 | 66.82 | 33.18 |
| DEF + H$_2$O | 321.0 | 36.06 | 42.21 | 22.51 | 72.69 | 27.31 |
| DMF + H$_2$O | 326.4 | 34.74 | 37.07 | 22.64 | 74.17 | 25.83 |

[a]The solution enthalpy of DCBNT.
[b]The solution enthalpy of DCBNT.
[c]The Gibbs free energy for the solution process of DCBNT.
[d]The relative contributions by enthalpy towards the solution process.
[e]The relative contributions by entropy towards the solution process under the experimental conditions.

that it is an entropy-driven dissolution process. Moreover, by comparing %$\zeta_H$ with %$\zeta_{TS}$, it can be concluded that the dissolution enthalpy is the main contributor of Gibbs free energy in the dissolution process of DCBNT, because all values of %$\zeta_H$ are ≥62.98%. In addition, $\Delta G_{sol}$ represents the minimum energy that is required to dissolve DCBNT under the experimental conditions. As shown in tables 5 and 6, the $\Delta G_{sol}$ value in DMSO + H$_2$O and DEF + H$_2$O is higher than that in the corresponding pure solvents, which is exactly the opposite in BL. So, the solubility of DCBNT is better in DMSO and DEF than in their binary solvents, but is better in BL + H$_2$O than in BL.

# 5. Conclusion

The solubility data of DCBNT in pure and binary solvents were measured at different temperatures from 290 to 360 K by the dynamic method. We can make the following conclusions: (i) the solubilities of DCBNT in all solutions increased with an increasing temperature; (ii) the solubility of DCBNT in DMSO is nearly 100 times higher than that of water and almost insoluble in DMF, methanol, ethanol, acetone, chloroform, dioxane, acetonitrile and trichloromethane, and the solubility of DCBNT in pure solvents is not only related to the polarity of solvent, but also related to other factors; (iii) the solubility data could be successfully correlated using the modified Apelblat model, the Yaws model and the polynomial model, and the fitting result of the three models is basically the same; and (iv) the thermodynamic properties for the solution process including Gibbs energy, dissolution enthalpy and the dissolution entropy were obtained by the Apelblat analysis and the standard Gibbs free energy shows that the dissolving process of DCBNT in all of these solvents is endothermic, and the enthalpy is a main contributor to the dissolution process of DCBNT.

Data accessibility. All the data in this investigation have been reported in the paper and are freely available.

Authors' contributions. J.R., D.C., Y.Y. and H.L. participated in all procedures including the design of the study. J.R. made substantial contributions to the acquisition and analysis of data. J.R. and D.C. drafted and revised the article. Y.Y. and H.L. gave final approval for publication and agreed to be accountable for all aspects of the work.

Competing interests. The authors declare no competing financial interest.

Funding. This work was supported by the National Natural Science Foundation of China (grant nos. U13301067 and 11672273).

Acknowledgements. We thank Prof. Guijuan Fan (Institute of Chemical Materials, China Academy of Engineering Physics, China) for her assistance with the raw materials.

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
