## [Reviewer comments · Royal Society Open Science]

Review History

RSOS-190728.R0 (Original submission)

Review form: Reviewer 1

Is the manuscript scientifically sound in its present form?

Yes

Are the interpretations and conclusions justified by the results?

Yes

Is the language acceptable?

Yes

Do you have any ethical concerns with this paper?

No

Have you any concerns about statistical analyses in this paper?

No

Recommendation?

Accept with minor revision (please list in comments)

Comments to the Author(s)

This work can be published in the journal when the corrections given below are made.

-Equations 1 and 2 are known equations. These can be replaced by a general Equation (Page 4, line 38 and Page 5, line 12), similar equations do not need to be rewritten.

- Only T must be substituted for T / K given in equations 3,4 and 5 (unless K is a constant or a variable) (Page 5 line 43 (equation 3); Page 5 line 10 (equation 4); Page 5 line 32 (equation 5)

Review form: Reviewer 2**Is the manuscript scientifically sound in its present form?**

Yes

Are the interpretations and conclusions justified by the results?

Yes

Is the language acceptable?

No

Do you have any ethical concerns with this paper?

No

Have you any concerns about statistical analyses in this paper?

No

Recommendation?

Accept with minor revision (please list in comments)

Comments to the Author(s)

The manuscript (No. RSOS-190728) has obtained the solubility data of an energetic ionic salt, DCBNT, in the different pure solvents and binary solvents by dynamic method, and the experiment data were correlated by three theoretical solubility models. The dissolution process was investigated by calculating the dissolution enthalpy, dissolution entropy, and Gibbs free energy. This work gave a large amount of solubility data for DCBNT and can help solvent selection for the recrystallization of DCBNT. So, I recommended it to publish in Royal Society Open Science after minor revision.

1. There are some grammatical mistakes in the whole manuscript, such as, inconsistent of subject and predicate, "Results in all the solvents was positively....." in the second sentence of abstract, "The solubility data of DCBNT was measured by....." in the first sentence of 2.2 section, "The solubility of DCBNT in all the solvents were tested" in the first sentence of left column in page 5, et al; lacking conjunction between sentences, "solubility of DCBNT in most solvents is almost insoluble, these solvents include DMF, methanol, ethanol, acetone, chloroform, dioxane, acetonitrile and ethyl acetate" in the first sentence of 4.1 section in page 6, "solvents is DMSO>DEF>H2O>BL, by.....the solubility of DCBNT in H2O should higher than that in DMSO" in line 11-22, page 7. Here, I don't point out all grammatical mistakes, please revise them carefully.

2. Authors referred to “the solubility of DCBNT in acetone+H₂O is abnormal” in the right column in line 31-33, page 12, but did not give detail about the abnormal phenomenon. Besides, we can find that the order of DCBNT solubility in pure solvents is DMSO>DEF>H₂O>BL, but the order changes into BL+H₂O>DMSO+H₂O>DEF+H₂O>DMF+H₂O>ACN+H₂O. It is better if authors can provide some explanations for the abnormality and change rule.

Decision letter (RSOS-190728.R0)

23-Sep-2019

Dear Professor Ren:

Title: Solubility of dicarbohydrazide bis[3-(5-nitroimino-1,2,4-triazole)] in Common Pure Solvents and Binary Solvents at Different Temperatures
Manuscript ID: RSOS-190728

Thank you for submitting the above manuscript to Royal Society Open Science. On behalf of the Editors and the Royal Society of Chemistry, I am pleased to inform you that your manuscript will be accepted for publication in Royal Society Open Science subject to minor revision in accordance with the referee suggestions. Please find the reviewers' comments at the end of this email. I apologise that this has taken longer than usual.

The reviewers and handling editors have recommended publication, but also suggest some minor revisions to your manuscript. Therefore, I invite you to respond to the comments and revise your manuscript.

Because the schedule for publication is very tight, it is a condition of publication that you submit the revised version of your manuscript before 02-Oct-2019. Please note that the revision deadline will expire at 00.00am on this date. If you do not think you will be able to meet this date please let me know immediately.

- 1) A text file of the manuscript (tex, txt, rtf, docx or doc), references, tables (including captions) and figure captions. Do not upload a PDF as your "Main Document".
- 2) A separate electronic file of each figure (EPS or print-quality PDF preferred (either format should be produced directly from original creation package), or original software format)

- 3) Included a 100 word media summary of your paper when requested at submission. Please ensure you have entered correct contact details (email, institution and telephone) in your user account
- 4) Included the raw data to support the claims made in your paper. You can either include your data as electronic supplementary material or upload to a repository and include the relevant doi within your manuscript
- 5) All supplementary materials accompanying an accepted article will be treated as in their final form. Note that the Royal Society will neither edit nor typeset supplementary material and it will be hosted as provided. Please ensure that the supplementary material includes the paper details where possible (authors, article title, journal name).

Best wishes,
Dr Laura Smith
Publishing Editor, Journals

On behalf of the Subject Editor Professor Anthony Stace and the Associate Editor Professor Tobias Hertel.

RSC Subject Editor:
Comments to the Author:
(There are no comments.)

RSC Associate Editor:
Comments to the Author:
Figs 2, 3 and 4 need to be re-submitted.

Reviewer comments to Author:
Reviewer: 1

Comments to the Author(s)
This work can be published in the journal when the corrections given below are made.

-Equations 1 and 2 are known equations. These can be replaced by a general Equation (Page 4, line 38 and Page 5, line 12), similar equations do not need to be rewritten.

- Only T must be substituted for T / K given in equations 3,4 and 5 (unless K is a constant or a variable) (Page 5 line 43 (equation 3); Page 5 line 10 (equation 4); Page 5 line 32 (equation 5)

Reviewer: 2

Comments to the Author(s)

The manuscript (No. RSOS-190728) has obtained the solubility data of an energetic ionic salt, DCBNT, in the different pure solvents and binary solvents by dynamic method, and the experiment data were correlated by three theoretical solubility models. The dissolution process was investigated by calculating the dissolution enthalpy, dissolution entropy, and Gibbs free energy. This work gave a large amount of solubility data for DCBNT and can help solvent selection for the recrystallization of DCBNT. So, I recommended it to publish in Royal Society Open Science after minor revision.

1. There are some grammatical mistakes in the whole manuscript, such as, inconsistent of subject and predicate, "Results in all the solvents was positively....." in the second sentence of abstract, "The solubility data of DCBNT was measured by....." in the first sentence of 2.2 section, "The solubility of DCBNT in all the solvents were tested" in the first sentence of left column in page 5, et al; lacking conjunction between sentences, "solubility of DCBNT in most solvents is almost insoluble, these solvents include DMF, methanol, ethanol, acetone, chloroform, dioxane, acetonitrile and ethyl acetate" in the first sentence of 4.1 section in page 6, "solvents is DMSO>DEF>H2O>BL, by.....the solubility of DCBNT in H2O should higher than that in DMSO" in line 11-22, page 7. Here, I don't point out all grammatical mistakes, please revise them carefully.

2. Authors referred to "the solubility of DCBNT in acetone+H2O is abnormal" in the right column in line 31-33, page 12, but did not give detail about the abnormal phenomenon. Besides, we can find that the order of DCBNT solubility in pure solvents is DMSO>DEF>H2O>BL, but the order changes into BL+H2O>DMSO+H2O>DEF+H2O>DMF+H2O>ACN+H2O. It is better if authors can provide some explanations for the abnormality and change rule.

Author's Response to Decision Letter for (RSOS-190728.R0)

See Appendix A.

Decision letter (RSOS-190728.R1)

07-Oct-2019

Dear Professor Ren:

Title: Solubility of dicarbohydrazide bis[3-(5-nitroimino-1,2,4-triazole)] in Common Pure Solvents and Binary Solvents at Different Temperatures

Manuscript ID: RSOS-190728.R1

It is a pleasure to accept your manuscript in its current form for publication in Royal Society Open Science. The chemistry content of Royal Society Open Science is published in collaboration with the Royal Society of Chemistry.

On behalf of the Subject Editor Professor Anthony Stace and the Associate Editor Professor Tobias Hertel.

RSC Associate Editor
Comments to the Author:
Authors appear to have addressed the criticism of reviewers 1 and 2.

Reviewer(s)' Comments to Author:

Appendix A

Responses:

For Referee 1

Comment one

Equations 1 and 2 are known equations. These can be replaced by a general Equation (Page 4, line 38 and Page 5, line 12), similar equations do not need to be rewritten.

Respond:

Thank you sincerely for your prompt. Equations 1 and 2 are similar, but the meanings are different. Equation 1 represents the solubility of DCBNT in pure solvent, and Equation 2 is the calculation of the solubility of DCBNT in binary solvents. If we do not give specific calculation equations, it may cause confusion to readers. So, we think it is necessary to hold it.

Comment two

Only T must be substituted for T / K given in equations 3,4 and 5 (unless K is a constant or a variable) (Page 5 line 43 (equation 3); Page 5 line 10 (equation 4); Page 5 line 32 (equation 5))

Respond:

We should apologize for the wrong expression. We have checked this kind of mistakes carefully and all errors have been amended in the revised paper.

For Referee 2

Comment one

There are some grammatical mistakes in the whole manuscript, such as, inconsistent of subject and predicate, “Results in all the solvents was positively.....” in the second sentence of abstract, “The solubility data of DCBNT was measured by.....” in the first sentence of 2.2 section, “The solubility of DCBNT in all the solvents were tested” in the first sentence of left column in page 5, et al; lacking conjunction between sentences, “solubility of DCBNT in most solvents is almost insoluble, these solvents include DMF, methanol, ethanol, acetone, chloroform, dioxane, acetonitrile and ethyl acetate” in the first sentence of 4.1 section in page 6, “solvents is DMSO>DEF>H2O>BL, by.....the solubility of DCBNT in H2O should higher than that in DMSO” in line 11-22, page 7. Here, I don't point out all grammatical mistakes, please revise them carefully.

Respond:

We apologize for many errors about punctuation mark and grammar in this paper and these main mistakes have been emended carefully.

Comment two

Authors referred to “the solubility of DCBNT in acetone+H₂O is abnormal” in the right column in line 31-33, page 12, but did not give detail about the abnormal phenomenon. Besides, we can find that the order of DCBNT solubility in pure

solvents is DMSO>DEF>H₂O>BL, but the order changes into BL+H₂O>DMSO+H₂O> DEF+H₂O>DMF+H₂O>ACN+H₂O. It is better if authors can provide some explanations for the abnormality and change rule.

Respond:

a. “Authors referred to “the solubility of DCBNT in acetone+H₂O is abnormal” in the right column in line 31-33, page 12, but did not give detail about the abnormal phenomenon.”

For this problem, after many experiments on the solubility of DCBNT in acetone+H₂O, it is found that the experimental data fluctuates greatly, we suspect that the possible cause is that acetone evaporates too quickly when applying our apparatus (CrystalSCAN), and the experimental results have no referential significance. Therefore, we only give " the solubility of DCBNT in acetone+H₂O is abnormal " in this manuscript in order to remind users to be cautious when using acetone+H₂O in the process of studying DCBNT.

b. “we can find that the order of DCBNT solubility in pure solvents is DMSO>DEF>H₂O>BL, but the order changes into BL+H₂O>DMSO+H₂O> DEF+H₂O>DMF+H₂O>ACN+H₂O.”

For this problem, this abnormality might be related to the solubilization of solvents. According to literatures, in “Solubilization of Steroids by Multiple co-Solvent Systems”¹, the author mentioned that in order to improve the solubility of many poorly soluble materials, scientists often incorporate one or more co-solvents with distilled water to overcome the poor aqueous solubility. As Jeffrey W. Millard² said, in several established methods for increasing the equilibrium solubility of non-polar drugs in aqueous vehicles, co-solvency, the addition of water miscible solvents to an aqueous system, is one of the oldest, most powerful, and most popular of these. In addition, in a study by R. K. Maheshwari³, he found that ethanol has weaker solubilizing power for frusemide. The addition of 15% w/v niacinamide (a solubilizer) in ethanol showed very good solubility of frusemide (about three-fold enhancement). After various experiments on solubilization, the author is of the opinion that all substances whether liquids, gases or solids possess solubilizing powers. Neelam Seedher et al⁴ believe that the greater the difference between the polarity of the 2 solvents in a given mixed solvent is, the greater the solubilization power will be. However, in a given mixed-solvent system, the solubilization power could not be related to the polarity of the drugs. In this work, we choose H₂O + related solvents to form binary solvents to dissolve DCBNT. The original intention is to improve the solubility of DCBNT. As for the interesting phenomenon of “the order of DCBNT solubility in pure solvents is DMSO>DEF>H₂O>BL, but the order changes into BL+H₂O>DMSO+H₂O> DEF+H₂O>DMF+H₂O>ACN+H₂O”, as stated in the literature, we hypothesized that the solubility of DCBNT increased in H₂O+BL mixed solvents with BL as a cosolvent or H₂O as a cosolvent compared to pure solvents. For the real reason of this phenomenon, we would like deeply to study in the future work, so we regret that we cannot give a precise explanation for now.

1. Chien, Y. W., Lambert, H. J., Solubilization of Steroids by Multiple co-Solvent Systems. *Chemical and Pharmaceutical Bulletin* **1975**, 23 (5), 1085-1090.
2. Millard, J. W., Alvarez-Núñez, F. A.; Yalkowsky, S. H., Solubilization by cosolvents. *International Journal of Pharmaceutics* **2002**, 245 (1-2), 153-166.
3. Maheshwari, R. K., Potentiation of solvent character by mixed-solvency concept :A novel concept of solubilization. *Journal of Pharmacy Research* **2010**, 3 (2), 411-413.
4. Seedher, N., Bhatia, S., Solubility Enhancement of Cox-2 Inhibitors Using Various Solvent Systems. *AAPS PharmSciTech* **2003**, 4 (3), 36-44.